# Insights and Evidence on Energy Retrofitting Practices in Rural Areas: Systematic Literature Review (2012–2023)

**Ahmed Abouaiana** * and **Alessandra Battisti**

Department of Planning, Design, and Technology of Architecture, Sapienza, University of Rome, Via Flaminia 72, 00196 Rome, Italy; alessandra.battisti@uniroma1.it
* Correspondence: abouaiana.ahmed@gmail.com; Tel.: +2-2011-1225-8628

**Abstract:** Rural commons face extraordinary challenges like fragility and sensitivity due to climate change. Retrofitting rural built environments affords benefits that could overcome these challenges and support sustainable development. However, notwithstanding the vast energy retrofitting interventions available, the associated aspects require investigation, particularly in distinct rural contexts with all their valuable, cultural, and historical inheritance. Hence, this study aimed to examine energy retrofitting practices in rural settlements worldwide over a decade to diagnose the goals that are being undertaken, stakeholder engagement, and finally, the bi-correlation between rural contexts and interventions, and retrofitting contributions to valorizing the place's identity. This study is a systematic literature review (SLR) considering the items of the PRISMA checklist (Preferred Reporting Items for Systematic Reviews and Meta-Analyses). An SLR of published peer-reviewed studies between January 2012 and March 2023 in 16 electronic databases in all available languages, using a combination of seven keywords within three domains, was conducted. The initial search resulted in 397; after applying the inclusion/exclusion criteria, there were 60 eligible articles. The academic progress and tendencies in the energy retrofitting domain of rural built environments are discussed and summarized into four major thematic classifications (energy efficiency strategies, energy efficiency planning, policy evaluation, and occupant behavior). Briefly, rural buildings lack energy-saving designs. Simulation tools are essential; however, they should be calibrated with on-site conditions, showing the reasons for selecting the applied retrofitting measures and correlation with the surrounding context. Successful implementation requires cross-disciplinary collaboration, engaging decision makers, and providing energy education for the local community. Regulations should include micro-context-specific environmental performance indicators. These insights could help map out future academic pursuits and help the stakeholders better understand their nature. Simultaneously, this study assists early-stage researchers in conducting systematic literature reviews utilizing different tools. However, the SLR protocol may have limited findings due to the specific search terms used, so the authors believe the more the literature search scope is broadened, the more discoveries could be made.

**Keywords:** qualitative approach; energy efficiency; rural commons; evidence-based practices; content analysis

## 1. Introduction

Traditional rural settlements emerged vernacularly (without architects [1]) or with architects and planners, while respecting the rural identity [2]. They have a distinguished architectural typology and urban materials through transferred knowledge between generations, to achieve dwellers' satisfaction. They are characterized by a high population density, relying on economic activities using the integration of primitive small-scale and modern techniques. Dwellers can solve problems, such as mobility, energy efficiency, and efficient use of space in their ways [3]. Rural areas play a crucial role in our daily lives by supporting the production of food and raw materials, offering recreational opportunities,

contributing to our overall well-being and environmental health, enhancing ecosystem services, and providing aesthetic value. They comprise the vast majority of the global territories and host half of the population in developing countries and one-third of the population in EU countries.

Nowadays, rural areas have undergone significant changes since the mid-20th century due to many reasons, like industrialization [4], philosophical shifts [5], and socioeconomic aspects [6], that produced modern built environment patterns. that produced modern built environment patterns, shifted the productive rural nature to consumerism which led to the consumption of natural resources and increases the demand for energy and water resources, all irreversible adverse impacts on the environment, making them confront distinct environmental challenges due to climate change as one of the most fragile areas [7–10].

Energy has played a crucial role in human civilization, but the energy industry is a major contributor to greenhouse gas emissions (e.g., nearly 80% in the EU). Scholars have highlighted the negative effects of this sector on the environment and emphasized its crucial role in achieving sustainable development goals (goal 7: supply affordable, sustainable, and reliable energy to all by 2030) [11]. In this context, the global community has intensified its efforts since the Paris Agreement in 2015 (an international treaty on climate change) to fulfill, at the national scale, the commitment to mitigate climate effects, particularly in rural contexts, where energy is a decisive factor [12]. For instance, the European Commission aims to support sustainable rural development through a number of initiatives in the framework of the European Green Deal [13]; rural specificity of a country is an essential characteristic influencing energy poverty [14]. The current policy in the United States of America (USA) is to finance Rural Clean Energy Initiative projects that focus on clean energy transitions [15]. This is regardless of the current global energy crisis due to shared global challenges, such as the Russian–Ukrainian conflict, which brings into mind the energy crisis that resulted in the aftermath of the Egyptian–Israeli war in the 1970s.

The building sector is a predominant aspect of the built environment, and is responsible for about 40% of global energy consumption. It has a substantial carbon footprint regarding the associated greenhouse gas emissions (GHG) from electricity consumption and anthropological activities. Energy retrofitting refers to adapting the latest technologies or features to obsolete systems [16]. Therefore, building energy retrofitting and utilizing existing clean energy solutions in rural areas can play a pivotal role in mitigating rural commons' challenges, such as meeting climate change goals [17], achieving low-carbon transitions [18], fulfilling sustainable development objectives [19], decreasing natural resource exploitation [20], and enhancing the quality of life [21].

Energy retrofitting practices are characterized by intricacy, unlike any domains, which was confirmed in the past two decades [22] and by a recent study [23]. This was through the efforts of the scholars, representing a significant increase in the scholarly publications in the field within this period [24]. Numerous studies defined this complexity in obstacles accompanying energy retrofitting domains related to different perspectives, like policies, user behavior and culture, techno-economic factors, and business engagement [25–27]. Moreover, the nature of the process requires cross-disciplinary collaboration among various stakeholders, as emphasized in Sibilla and Kurul's systematic review titled "Transdisciplinarity in energy retrofit" [28].

Briefly, cross-disciplinarity has different forms. Multi-disciplinarity: coherence of the conceptual frameworks associated with various disciplines, focusing on a particular topic from different angles (working together but without significant contributions). Intra-disciplinary: combines two or more diverse sub-fields under the major one, concentrating participants' efforts to derive concepts. Inter-disciplinarity: builds a mutual framework for the collaborating disciplines under the same field to create a synthesis, working on common questions to achieve shared results (the results are more than the sum of each discipline). Trans-disciplinarity: multi-level coordination incorporating academic researchers from diverse fields with non-academic participants and professional practices to flourish the

epistemology and theory for knowledge generation [29–31]. Figure 1 visualizes the Latin origin of the concepts for a better understanding.

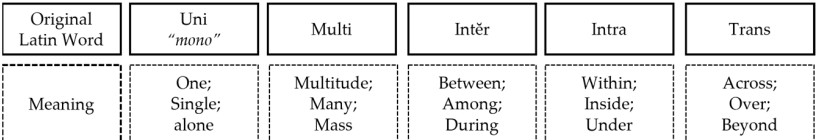

**Figure 1.** The Latin origin of the cross-disciplinarity forms.

Rural commons have specific socio-cultural values, which is higher than that in urban contexts [32], which can be represented by, the strong bond between residents and their home [33], regardless of the physical identity of the rural buildings (despite their steady transformations [34]). Consequently, many studies have emphasized the necessity of preserving the cultural and natural values of rural places while developing them [35–38], including energy retrofitting.

Therefore, this study aimed to identify the retrofitting practices for energy efficiency in rural settlements in the past decade to highlight the progress in academic research, synthesize the evidence into insights [39], and provide a meticulous, holistic summary through evidence-based practices, such as a systematic literature review (SLR) [40,41]. SLR is a holistic search for relevant scientific research on the investigated topic using structured methodologies [42]. We aim to provide an explicit model of the status of global research on rural built environments beyond the technical interventions and the numerical results (energy saving quantifying). We hypothesize that rural contexts differ from others, represented by the reciprocal correlation between interventions and the surrounding contexts. This can be addressed by answering three research questions (RQs):

- (RQ1) What are the thematic classifications of the activities undertaken within these practices, considering cross-disciplinarity and stakeholder engagement?
- (RQ2) What is the relationship between rural contexts and interventions?
- (RQ3) To what extent do the interventions contribute to valorizing the rural identity?

## 2. Materials and Methods

The aim was accomplished and the research questions were addressed by conducting an SLR. We used the PRISMA (Preferred Reporting Items for Systematic Reviews and Meta-Analyses) checklist [43], which emerged in 2009 [44]. The checklist was employed to identify the research question, search keywords that meet the objectives, and synthesize the qualitative results.

The proposed SLR steps were revised based on references [45,46]. The SLR was executed based on the modified protocol, using explicit inclusion and exclusion criteria (Section 2.1) and employing the relevant keywords (retrofitting rural built environments) (Section 2.2) which led to a comprehensive examination of the literature to identify retrofitting factors globally. Figure 2 depicts the SLR process and structure.

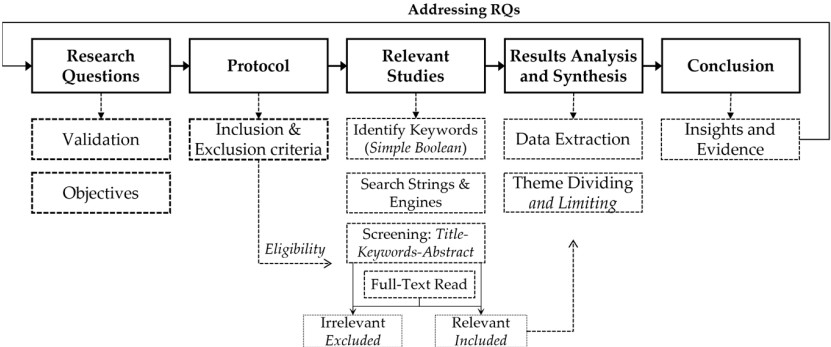

**Figure 2.** The SLR procedure and structure.

### *2.1. SLR Protocol: Inclusion and Exclusion Criteria*

The search selected peer-reviewed articles in journals and conferences to ensure their quality. They should be primary sources and published within the last decade (between 2012 and March 2023) in all available languages and electronic databases. Books, grey literature (e.g., M.Sc. and Ph.D. theses), and secondary sources were excluded.

Meanwhile, the articles should address the specific problem and answer at least one research question. Namely, they had to report energy efficiency improvement strategies or practices in the buildings or surrounding environment or discuss retrofitting policies in rural contexts. Conversely, the articles that distinctly discuss non-rural contexts (i.e., urban and suburban areas) or undefined contexts or provide generic guidelines or evaluated policies were excluded. Table 1 summarizes the exclusion and inclusion criteria.

**Table 1.** The inclusion and exclusion criteria for the SLR.

| Factor | Inclusion Criteria | Exclusion Criteria |
|---|---|---|
| Document Type | Peer-reviewed journal articles; Peer-reviewed conference articles; Primary research. | Grey literature (e.g., M.Sc. and Ph.D. theses); Books and book chapters; Secondary research. |
| Year Range | Between January 2012–March 2023 | Before January 2012 and after February 2023 |
| Ultimate context and intimate context | All kinds of rural settlements (e.g., historical and abandoned villages); Discusses retrofitting (policies/practices) in generic contexts, including rural ones. | Not rural or undefined context (e.g., cities, urban or suburban contexts) |
| Relevance to the objectives | The articles address "retrofitting" for energy efficiency and answer one or more research question(s). | The article discusses a specific topic not relevant to the research questions. |
| Language | All available languages | Not applicable (n/a) |
| Research topic | Retrofitting built environment; Case studies and best practices; Review policies or energy efficiency programs that specified rural contexts; | Review policies or energy efficiency programs with general guidelines that are not specific to rural contexts; Specific topic (e.g., material development); Does not include retrofitting or any synonyms. |

### *2.2. Research Strategy*

The research context's keywords were specified to construct the search string. Three strings of search keywords were used. The first term is "retrofitting" and its synonyms "refurbishment" and "renovation" as they are three commonly used expressions to describe physical changes executed to improve existing buildings or built environments [47]; the asterisk is used to give a different form of the keyword, for instance (renovat* may indicate renovation, renovated, renovated, or renovating). The second term is "energy efficiency" or "energy saving" which is the core of the research objective. The third term is "rural." Meanwhile, in order to broaden the obtained results, related built environment aspects like buildings and the context classifications (e.g., village, settlement, community) were excluded. However, these aspects were extracted during the review process. The search string implemented using simple Boolean (AND and OR) logic:

**(**retrofit* OR refurbishment OR renovat***) AND (**"energy efficiency" OR "energy saving" OR "energy-efficien*"**) AND** rural

This study used the Rome Digital Library System of Sapienza University—SBS (Discovery Sapienza) powered by EBSCOhost (https://web.uniroma1.it/sbs/en/discovery sapienza, accessed on 22 June 2023), as well as EBSCOhost which is one of the commonly employed databases by scholars in different disciplines [48]. It searches various electronic sources simultaneously using the maximum number of keywords; of note is that the platform can be utilized without registration, and only a few additional features are exclusive to users with an institutional email, like saving the research result in folders under the account.

### 2.3. Results and Study Selections

The initial search process on SBS was conducted between January 2012 and March 2023, applying the Peer Review filter; 353 studies were found, in 16 electronic databases, Figure 3, all available in English. The SBS engine auto-detected duplications; these were removed and the number decreased to 221 studies. These were exported in (RIS) format (available in Supplementary File S1) to a reference manager software Zotero V.6 to check for further duplications, omitting 53 more articles, leaving 168 studies which were exported for screening using the Rayyan online platform (https://www.rayyan.ai, accessed on 15 March 2023). At this stage, one duplicated article was eliminated (provided in two languages).

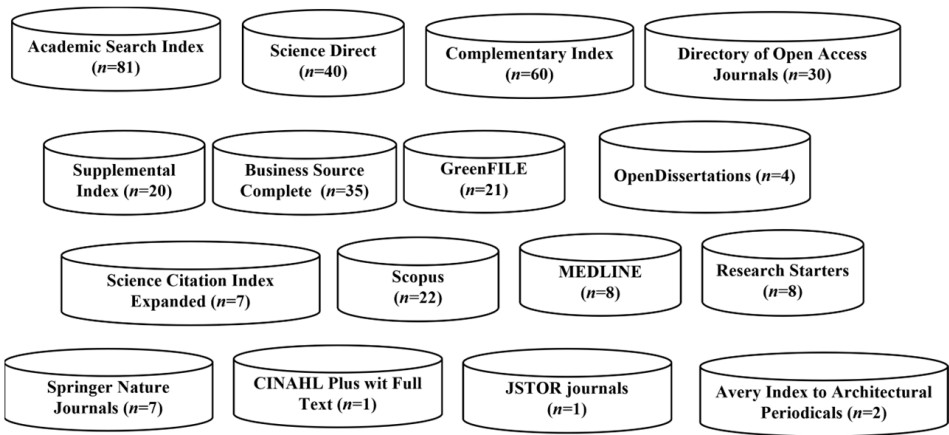

**Figure 3.** The number of publications from each database.

Rayyan is a common tool for SLR in many domains [49], it allows users to review each paper (displaying the title, abstract, keywords, publisher, and authors) and add notes and PDF files. The interface shows a summary, for instance, of screening time and the number of sessions. Each paper can be labeled and selected to be included, excluded, or labelled as maybe (to be decided later). It enables collaboration among the reviewer team which accelerates the study selection process and decreases the screening time by 40% compared to similar tools [50].

The evaluation process was implemented in two steps. Firstly, the initial screening process of the title, keywords, and abstract was implemented via the Rayyan platform. The screening process excluded 92 articles that could not be related to the research questions. The remaining articles (75) qualified for the second step. It should be noted that if one author or both authors had reservations about approving the study, the article was accepting for the next step (full-paper analysis).

In the second phase, the papers were downloaded and read one by one. Some manuscripts had limited access, but the authors could download them through institutional accounts. In this second evaluation process, 18 papers were excluded. Figure 4 illustrates the PRISMA flowchart of the selection process.

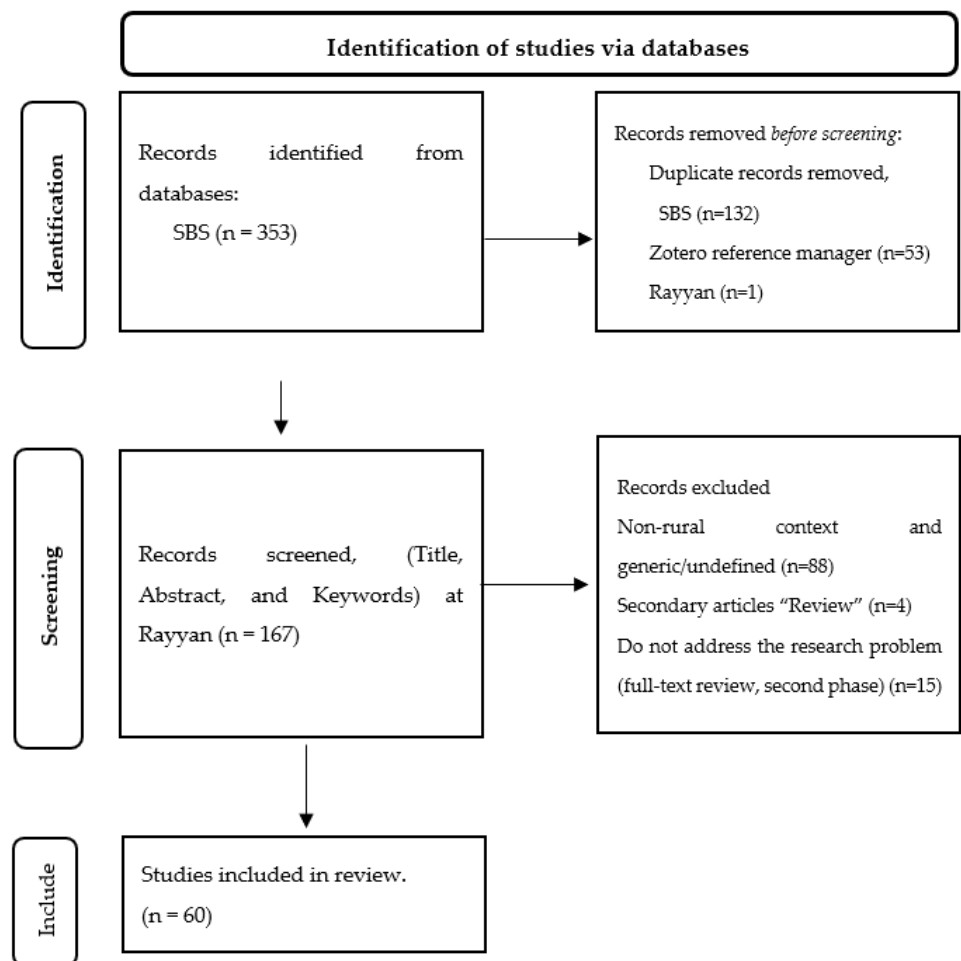

**Figure 4.** The PRISMA flow diagram of the selection of articles.

### 2.4. Data Extraction

After selecting the eligible articles, the authors developed a data extraction form in an Excel sheet (available in Supplementary File S2). The extracted data are divided as follows:

- The bibliographical data, authors' names, numbers, citation score, and keywords (Section 3.1);
- Authors' affiliations, affiliation countries, and cross-disciplinarity (Section 3.2);
- Characteristics of the geographical and micro-contexts (Section 3.3);
- Scope of analysis, focus, and theme classifications (Section 3.4);
- Result synthesis (Section 3.5).

In conclusion, the SLR resulted in 60 peer-reviewed papers (36% of the total articles). This occurred because of the rigid restriction of the eligibility criteria, which searched a combination of 7 keywords within three domains between January 2012 and March 2023. The included articles are shown in Table 2.

**Table 2.** The included SLR studies (in ascending order by year of publication).

| ID | Ref | Title |
|---|---|---|
| ID01 | [51] | State of the Irish housing stock—Modelling the heat losses of Ireland's existing detached rural housing stock & estimating the benefit of thermal retrofit measures on this stock |
| ID02 | [52] | Evaluating fuel poverty policy in Northern Ireland using a geographic approach |
| ID03 | [53] | Renovation alternatives to improve energy performance of historic rural houses in the Baltic Sea region |
| ID04 | [54] | Overview of rural building energy efficiency in China |

**Table 2.** *Cont.*

| ID | Ref | Title |
|----|-----|-------|
| ID05 | [55] | Azioni e strumenti per il recupero e la valorizzazione dell'architettura e del paesaggio rurale e montano |
| ID06 | [56] | A comprehensive analysis of building energy efficiency policies in China: status quo and development perspective |
| ID07 | [57] | Impact of Civil Envelope on Energy Consumption based on EnergyPlus |
| ID08 | [58] | Analysis on building energy performance of Tibetan traditional dwelling in cold rural area of Gannan |
| ID09 | [59] | Material flow accounting for an Irish rural community engaged in energy efficiency and renewable energy generation |
| ID10 | [60] | Retrofitting domestic appliances for PV powered DC Nano-grid and its impact on net zero energy homes in rural India |
| ID11 | [61] | An examination of energy efficiency retrofit depth in Ireland |
| ID12 | [62] | Role of Self-Efficacy in Reducing Residential Energy Usage |
| ID13 | [63] | Evaluation of refurbishment alternatives for an Italian vernacular building considering architectural heritage, energy efficiency and costs |
| ID14 | [64] | Analysis and Optimization on Energy Performance of a Rural House in Northern China Using Passive Retrofitting |
| ID15 | [65] | Effect of Building Roof Insulation Measures on Indoor Cooling and Energy Saving in Rural Areas in Chongqing |
| ID16 | [66] | Energy retrofit and environmental sustainability improvement of a historical farmhouse in Southern Italy |
| ID17 | [67] | Thermal comfort optimisation of vernacular rural buildings: passive solutions to retrofit a typical farmhouse in central Italy |
| ID18 | [68] | Redesign of a Rural Building in a Heritage Site in Italy: Towards the Net Zero Energy Target |
| ID19 | [69] | Analysis of Passive Energy-saving Retrofitting of Rural Residential Houses in Southern Anhui Province—A case in Hongcun |
| ID20 | [70] | Sustainability evaluation of retrofitting solutions for rural buildings through life cycle approach and multi-criteria analysis |
| ID21 | [71] | SWOT Analysis for the Promotion of Energy Efficiency in Rural Buildings: A Case Study of China |
| ID22 | [72] | Geometric classification method of rural residences at regional scale |
| ID23 | [73] | China's building stock estimation and energy intensity analysis |
| ID24 | [74] | An exploration about the Solar Energy Utilization and the Enclosure System Renovation for Rural Residential Buildings in Cold Areas of Northern China—Taking the rural residential renovation design in Zhujialin Village, Linyi as an example |
| ID25 | [75] | Residential energy transition and thermal efficiency in an arid environment of northeast Patagonia, Argentina |
| ID26 | [76] | Towards a cleaner domestic heating sector in China: Current situations, implementation strategies, and supporting measures |
| ID27 | [77] | Energy, carbon, and cost analysis of rural housing retrofit in different climates |
| ID28 | [78] | Indoor Temperature Improvement and Energy-Saving Renovations in Rural Houses of China's Cold Region—A Case Study of Shandong Province |
| ID29 | [79] | Facility Energy Management Application of HBIM for Historical Low-Carbon Communities: Design, Modelling and Operation Control of Geothermal Energy Retrofit in a Real Italian Case Study |
| ID30 | [80] | An Integrated HBIM Simulation Approach for Energy Retrofit of Historical Buildings Implemented in a Case Study of a Medieval Fortress in Italy |
| ID31 | [81] | Reduced biodiversity in modernized villages: A conflict between sustainable development goals |
| ID32 | [82] | The Economic Effects of New Patterns of Energy Efficiency and Heat Sources in Rural Single-Family Houses in Poland |
| ID33 | [83] | Preliminary Energy Evaluations for the Retrofit of Rural Protected Buildings in a Peripheral Context of Milan |
| ID34 | [84] | Heat consumption scenarios in the rural residential sector: the potential of heat pump-based demand-side management for sustainable heating |
| ID35 | [85] | Integrated assessment of the environmental and economic effects of "coal-to-gas conversion" project in rural areas of northern China. |
| ID36 | [86] | Renewable Energy Utilization in Rural Residential Housing: Economic and Environmental Facets |
| ID37 | [87] | The role of personal and environmental factors in rural homeowner decision to insulate; an example from Poland |
| ID38 | [88] | Active–passive combined energy-efficient retrofit of rural residence with non-benchmarked construction: A case study in Shandong province, China |
| ID39 | [89] | Life Cycle Carbon Emission Analyzing of Rural Residential Energy Efficiency Retrofit-A Case Study of Gansu province |
| ID40 | [90] | Framework for design and optimization of a retrofitted light industrial space with a renewable energy-assisted hydroponics facility in a rural northern Canadian community |

**Table 2.** *Cont.*

| ID | Ref | Title |
|---|---|---|
| ID41 | [91] | Life-Cycle Assessment of a Rural Terraced House: A Struggle with Sustainability of Building Renovations |
| ID42 | [92] | Evaluation of Rural Dwellings' Energy-Saving Retrofit with Adaptive Thermal Comfort Theory |
| ID43 | [93] | Retrofitting Rural Dwellings in Delta Region to Enhance Climate Change Mitigation in Egypt |
| ID44 | [94] | Theoretical Study on the Relationship of Building Thermal Insulation with Indoor Thermal Comfort Based on APMV Index and Energy Consumption of Rural Residential Buildings |
| ID45 | [95] | Passive Energy-Saving Optimal Design for Rural Residences of Hanzhong Region in Northwest China Based on Performance Simulation and Optimization Algorithm |
| ID46 | [96] | Analysis of Energy Performance and Integrated Optimization of Tubular Houses in Southern China Using Computational Simulation. |
| ID47 | [97] | Mitigating heat demand peaks in buildings in a highly renewable European energy system |
| ID48 | [98] | Evaluation of energy-saving retrofit projects of existing rural residential envelope structures from the perspective of rural residents: the Chinese case |
| ID49 | [99] | Net-zero energy retrofit of rural house in severe cold region based on passive insulation and BAPV technology |
| ID50 | [100] | Environment improvement and energy saving in Chinese rural housing based on the field study of thermal adaptability |
| ID51 | [101] | Evaluation of energy-saving retrofits for sunspace of rural residential buildings based on orthogonal experiment and entropy weight method |
| ID52 | [102] | Opportunities stemming from retrofitting low-resource East African dwellings by introducing passive cooling and daylighting measures |
| ID53 | [103] | Estimating the impact of energy efficiency on housing prices in Germany: Does regional disparity matter? |
| ID54 | [104] | Evolutionary Game Analysis of Energy-Saving Renovations of Existing Rural Residential Buildings from the Perspective of Stakeholders |
| ID55 | [105] | Exploring pathways of phasing out clean heating subsidies for rural residential buildings in China |
| ID56 | [106] | Energy Saving and Thermal Comfort Performance of Passive Retrofitting Measures for Traditional Rammed Earth House in Lingnan, China |
| ID57 | [107] | A study on influencing factors of optimum insulation thickness of exterior walls for rural traditional dwellings in northeast of Sichuan hills, China |
| ID58 | [108] | Green retrofit of existing residential buildings in China: An investigation on residents' perceptions |
| ID59 | [109] | Sustainable Energy Development and Climate Change Mitigation at the Local Level through the Lens of Renewable Energy: Evidence from Lithuanian Case Study |
| ID60 | [110] | Improving Building Envelope Efficiency Lowers Costs and Emissions from Rural Residential Heating in China |

## 3. Results and Analysis

### 3.1. Publications, Authors, Affiliations

The distribution of papers by year demonstrates a steady increase of about 9% each year in publication rate in the latest three years (Figure 5a). China contributed half of the studies on this subject. Studies in Italy came in second with 15%, followed by Poland, and the United Kingdom (UK) and Ireland with 7% and 5%, respectively. The remaining articles were implemented in 13 countries (6 in Europe, 2 in Asia, 2 in Africa, 1 in South America, and 2 in North America), Figure 5b.

A total number of 259 authors affiliated with 80 countries produced these articles. Chinese affiliations came in first place with 31%, followed by Italy at 11%, the United Kingdom (UK) and Ireland at 9%, the United States of America (USA) and Poland at 8% each, and 2% of each for Turkey, Sweden, the Netherlands, Germany, and Japan; the remaining countries had one affiliation each (Figure 5c).

Twelve authors have collaborated twice and produced eight articles: Grohmann, D., and Menconi, M.E. [67,70]; Cotana, F., Piselli, C., and Romanelli, J. [79,80]; Klepacka. A.M. [86,87]; and Arıcı, M., Jiang, W., Li, Q., Li, D., and Qi, H. [99,101]. As remarked from the timeline of publications, it seems these authors split/developed the same research line for two publications. The collaboration rate among authors was predominated by 2–5 authors, at a frequency of 35 times, followed by 6–10 authors that collaborated in 23% of the papers. Only one article was conducted by one author, showing the need for more multi- and inter-disciplinarity, supporting the findings of Sibilla and Kurul.

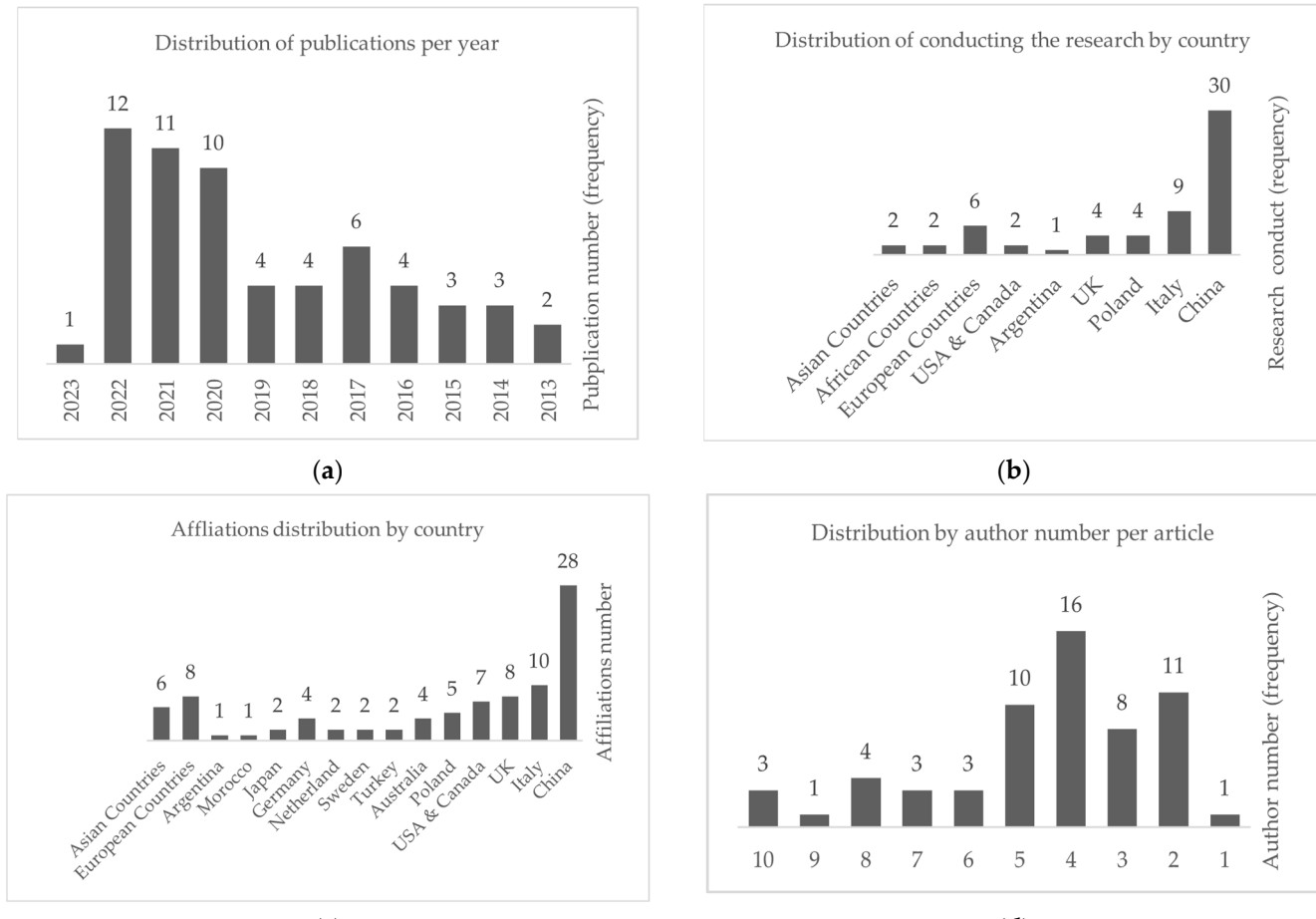

**Figure 5.** Quantitative analysis of included articles. (**a**) Distribution of articles by year; (**b**) distribution of research by country; (**c**) distribution by affiliations' country; (**d**) distribution by author count of paper.

The citations from three databases indicated a positive correlation between the year of publication and citation number, with more citations in later years (Figure 6a). The top five cited papers are [51,53,54,56,73], all published by Elsevier. In this logic, Huo et al.'s recent study (in 2019) [73] had the most citations, double that of the next highest cited paper in the same reference year (but no observed correlation between author count and citation numbers). Google Scholar (GS) had the highest citation number for all publications compared to the other two databases (Web of Science and Scopus); both almost completely overlapped with GS. The average overlap between Web of Science (WoS) and Scopus with GS was 73% and 63%, respectively, and WoS coincided with Scopus by two-thirds.

In terms of publisher (in Figure 6b) 53% of the articles were published in Journals by Elsevier (Figure 6c) between 2013 and 2022 and achieved a total citation of 954, 720, and 616 in GS, Scopus, and WoS, respectively. The average citation per paper, respectively, was 30, 23, and 19. MDPI's journals (Figure 6d) was second with 25% (published between 2017 and 2022) and achieved citation numbers of 184, 150, and 144 in GS, Scopus, and WoS, respectively, with an average of 12, 10, and 10 for each paper. This was followed by Springer with two articles. The remaining articles were distributed equally among ten publishers. In terms of journals, *Energies* was first with eight articles, mostly discussing energy efficiency planning and scenarios. In second, *Energy & Buildings* articles mainly discussed bottom-up retrofitting strategies. Next were *Energy for Sustainable Development*, *Energy Procedia*, and *Journal of Cleaner Production*, each with four articles, followed by *Buildings* and *Energy Policy* with three each.

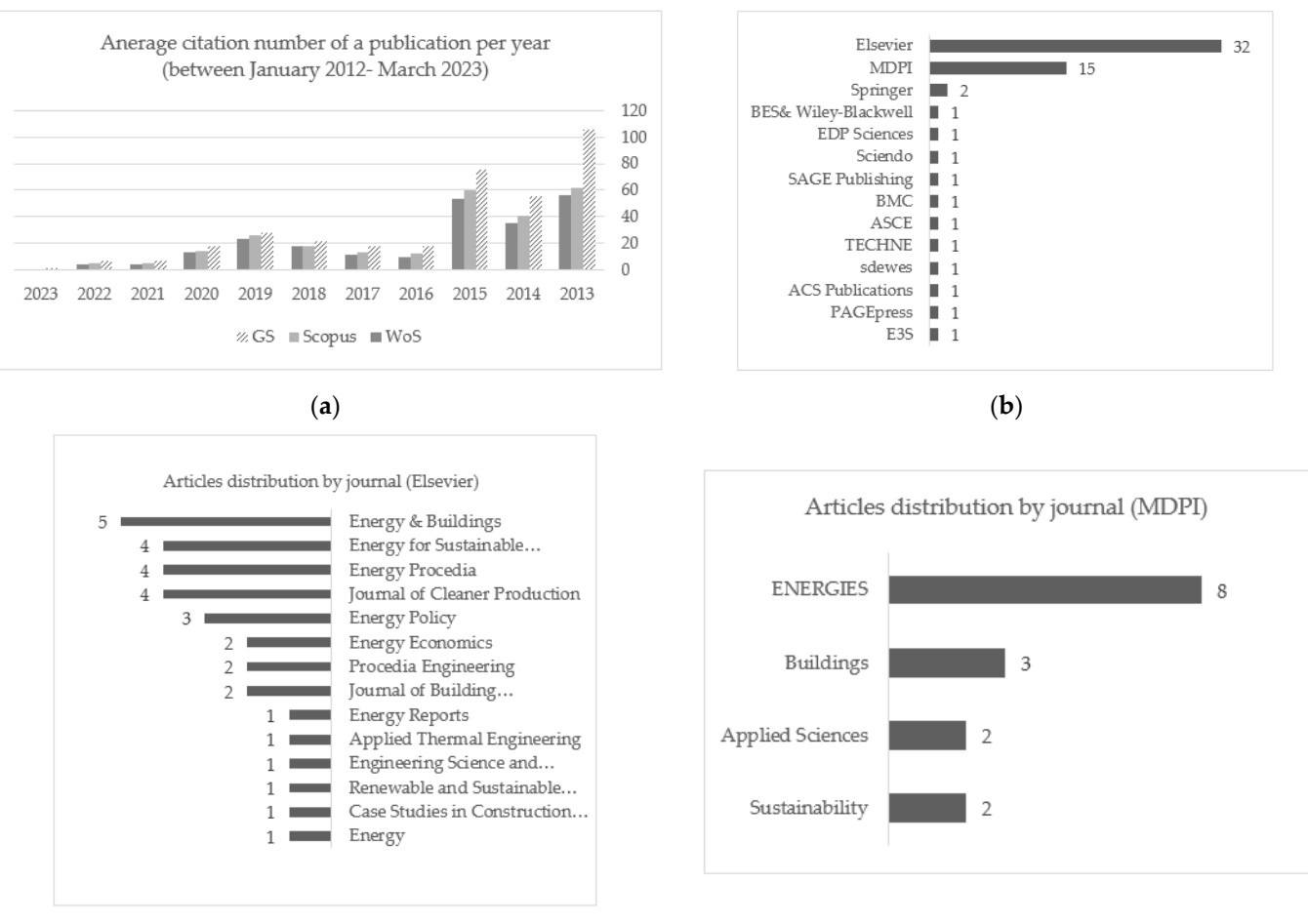

(**a**)

(**b**)

(**c**)

(**d**)

**Figure 6.** (**a**) Annual average citation per article for each database; (**b**) distribution of publishers; (**c**) distribution of journals published by Elsevier; (**d**) distribution of journals published by MDPI.

*3.2. Affiliations and Cross-Disciplinarity (Supplementary File S2)*

In terms of the affiliations for the included articles, almost all the papers were implemented by academic bodies and research centers; only the authors of five papers (9%) were affiliated with governmental bodies [61,66,76,104,105]. Additionally, 9% of papers included authors from the private sector in the domains of technology, e2nergy, and energy economics [56,80,89,98,102]. It was apparent that there was an absence of local communities and social contributions to the publications, apart from a few papers, that engaged the owner/landlords in early decision making [70]. The institution that produced the most articles was the University of Perugia in Italy with four articles, followed by the Polish Academy for Science in Poland with three papers, 12 institutions which produced a few papers, and the remaining institutions published one article each. Figure 7 shows the top institutions by author affiliation.

To identify the primary domains that produced the included studies, we examined the affiliated departments and then collected, classified, and grouped them under 15 domains, as seen in Table 3. The top contributions were architecture and planning at 38% of the total, followed by applied and environmental sciences at about 22%, the energy domain at 15%, and building and construction management and economics at 12%.

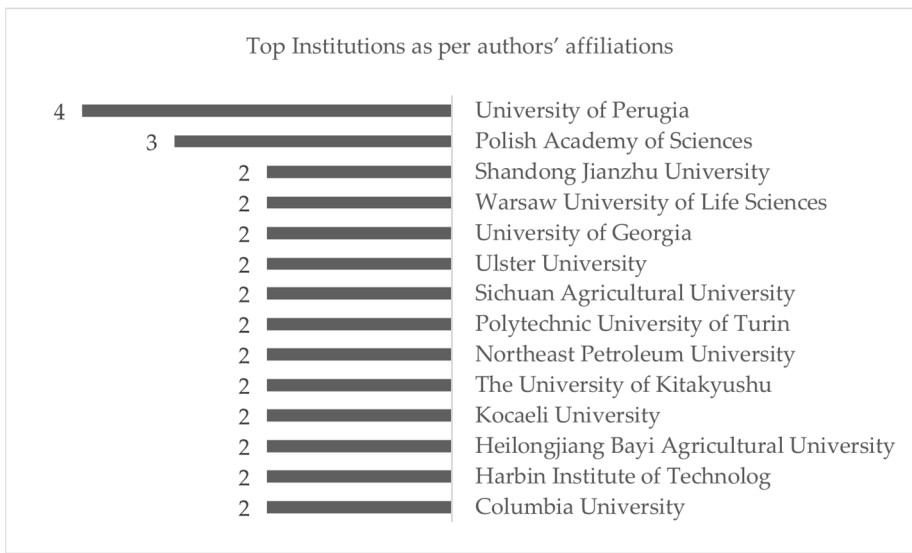

**Figure 7.** Most frequent institutions by author affiliation.

**Table 3.** Classification of the scientific domain of authors' departments extracted from the included papers (available in Supplementary Files S3 and S4).

| Discipline | Frequency | The Description Extracted from the Department's Names |
|---|---|---|
| Architectural and urban/rural planning | 23 | Architecture, urban or rural planning technology for architecture, architecture and built environment, landscape architecture, and maritime architecture |
| Applied Sciences | 14 | Civil engineering, chemical engineering, and mechanical engineering |
| Environmental Sciences | 12 | Environmental engineering and hydropower engineering |
| Energy | 9 | Energy technology, energy efficiency economics, and energy |
| Economics | 7 | Finance |
| Building and Construction | 7 | Building physics, building and real state, and construction management |
| Agricultural science | 5 | Rural and agriculture development and agricultural and applied economics |
| Political science | 3 | Public and international affairs and environmental policies |
| Technology | 3 | Technology development |
| Biology | 2 | Biological and geoenvironmental technologies and zoology |
| Business and management | 2 | Management and economics, sustainable development, and smart decision making |
| Computer Science | 2 | |
| Art History | 1 | Art and history (conservation) |
| Psychology | 1 | |
| Mathematics | 1 | |

The collaboration level in the included studies varied. Half of the articles were conducted by one discipline, predominated by architecture and planning. Meanwhile, 42% of the studies were produced by two domains, mainly the combination of architecture and urban/rural planning, followed by architecture and applied engineering (4 articles), architectural and environmental sciences (3 articles), and architecture and planning with building and construction (2 papers). Figure 8 maps out the cross-disciplinarity of the included papers.

The total number of keywords, as stated in the papers, was 297 (Figure 9), with an average of 5/article. As seen in the word cloud, the most frequently used keyword was energy efficiency, and in second was building energy efficiency, energy consumption, retrofit, and China. The third group included energy retrofit, rural residence, sustainability, and multi-objective optimization. The fourth group included cultural heritage, historical

buildings, thermal insulation, rural dwellings, renewable energy, and economic analysis, which indicates an observed correlation between the SLR protocol and keywords.

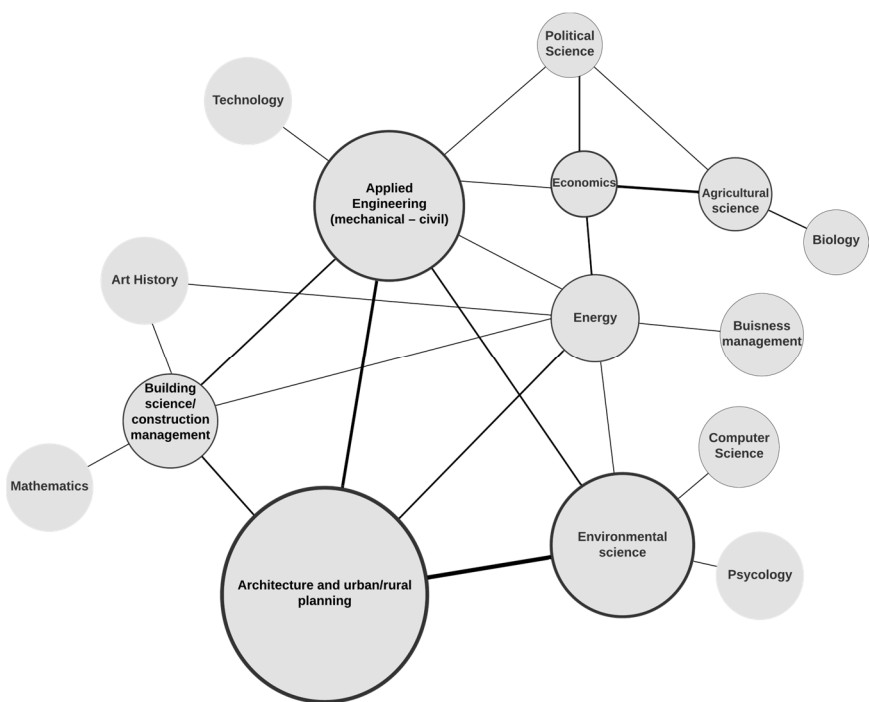

**Figure 8.** Visual map illustrating the connectivity and strength of collaboration among the different domains in the included studies (the frequencies of each discipline and the cross-disciplinary collaborations are available in Supplementary File S4).

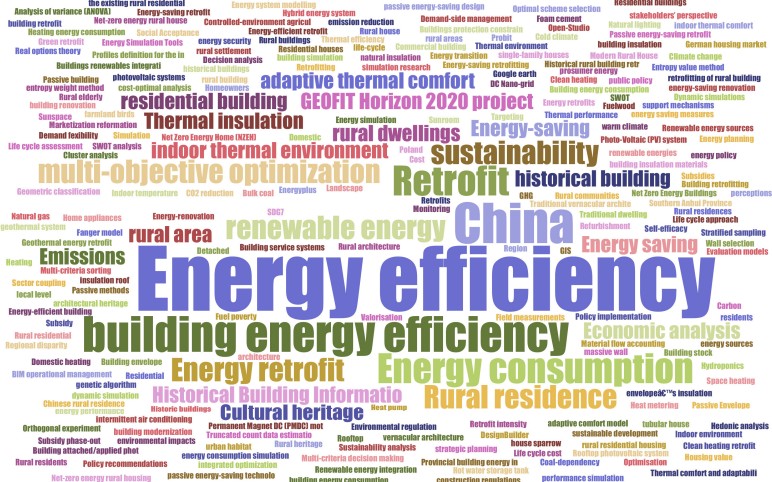

**Figure 9.** A visual summary of the keywords provided by the included studies (available in Supplementary File S3), generated by https://www.jasondavies.com/wordcloud/, accessed on 15 May 2023.

### 3.3. Micro-Context Patterns

In terms of micro-context, all the articles have focused on only rural contexts, apart from a few papers that evaluated energy policies in generic contexts (urban and rural) that clearly indicated applications to rural contexts (Figure 10). Approximately 70% did not define a particular context or settlement pattern; 7% indicated agriculture-based settlements; 12% applied to historical and listed buildings in rural areas, except for protected rural buildings in a city [83] and a rural building in an archeological park in a city [68]; 4% were conducted in remote and abandoned villages in India [60], China [69], and Argentina [75];

4% mentioned the adaptive reuse of a touristic building in Italy and China [66,107]; 2% were implemented in urban areas, namely, the countryside in a city [71]; and one article was implemented in mountainous areas [69]. Figure 10 summarizes the pattern characteristics of the investigated micro-context.

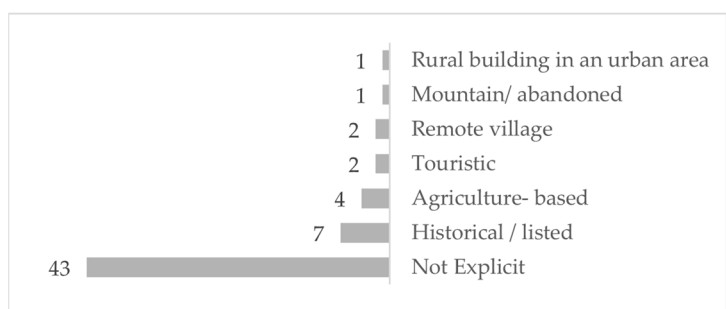

**Figure 10.** Characteristics and patterns of the examined context scope of the included articles.

### 3.4. Analysis Scope and Theme Classification

In terms of analysis scope, the articles have analyzed at least one or more aspects as the main aim of the paper (Table 4) which can be summarized as:

- One-factor analysis: Eighteen articles (30%) investigated one aspect, i.e., energy (energy efficiency and clean heating), and only one paper [67] examined retrofitting to attain thermal comfort in a traditional farmhouse building.
- Two-factor analysis: A total of 32 papers (53%) analyzed two aspects, namely, energy/cost (15 articles), energy/thermal comfort (6 articles), energy/carbon (4 articles), energy/valorization (2 articles), energy/human behavior (1 article), and carbon/cost (1 article).
- Three-factor analysis: Nine papers have provided multi-objective optimization, namely, energy–thermal comfort with valorization and air quality (three papers) and the others investigated energy, cost, and thermal comfort.
- Four-factor analysis: one paper discussed energy, cost, carbon, and air quality.

Energy analyses predominated in 97% of papers and overlapped with all factors apart from two articles that focused on retrofitting for achieving thermal comfort, vernacular dwellings for thermal comfort, and investigating carbon-associated energy saving and cost. Economic analyses (economic assessment, cost, and energy economics) came second with 40% and third was thermal comfort (17%). Only 7% clearly associated valorizing the traditional identity with the aims, and the same percentage considered indoor/outdoor air quality. Hydroponics farming was discussed as a retrofitting solution in one paper. Finally, the latest paper discussed an unusual approach led by biodiversity: investigating the impact of existing building retrofitting on birds. Table 4 summarizes the analyzed factors. Note that some papers examined different aspects as supportive elements to the main objective. For instance, reference [93] discussed users' perspectives on the implemented retrofitting, while the main goal was to quantify the potential energy saving.

The content review and analysis of the included studies led to their classification based on the central theme with each containing sub-themes. They can be classified into four categories: (1) providing energy efficiency strategies (57%), mainly quantifying potential energy savings of suggested retrofitting measures; (2) discussing energy efficiency planning (17%) to help guide the future for stakeholders via delivering energy efficiency and/or retrofitting guidelines for developing policies and supporting decision making [111]; (3) policy evaluation (18%) to assess the energy policies related to building retrofitting and reviewing national retrofitting schemes; and (4) discussing social and human behavioral aspects (8%), like the perception of the practices and willingness to retrofit.

**Table 4.** Analysis scope according to the aim of included studies.

| Analyzed Factors | Frequent | Energy | Economics/Cost | Thermal Comfort | Carbon and/or GHG | Air Quality | Valorization | Human Behavior | Hydroponics Farming | Biodiversity |
|---|---|---|---|---|---|---|---|---|---|---|
| One | 18 | x | | | | | | | | |
|  | 1 | | | x | | | | | | |
| Two | 16 | x | x | | | | | | | |
|  | 6 | x | | x | | | | | | |
|  | 2 | x | | | | | x | | | |
|  | 4 | x | | | x | | | | | |
|  | 1 | x | | | | | | | x | |
|  | 1 | x | | | | | | | | x |
|  | 1 | x | | | | | | x | | |
|  | 1 | | x | | x | | | | | |
| Three | 2 | x | x | | | x | | | | |
|  | 2 | x | x | | x | | | | | |
|  | 1 | x | | x | | x | | | | |
|  | 1 | x | x | | | | x | | | |
|  | 1 | x | | x | x | | | | | |
|  | 1 | x | x | x | | | | | | |
| Four | 1 | x | x | | | x | x | | | |
| Total | | 58 | 24 | 10 | 8 | 4 | 4 | 1 | 1 | 1 |

*3.5. Results Analysis and Synthesis*

A quantitative analysis has been provided, showing the characteristics of the included articles; based on this and the four main classifications, a qualitative analysis was used to synthesize the results. In line with the study hypotheses, each pattern affects/is affected by the retrofitting strategy to add value to the area and diagnose the main implemented activities beyond the in-depth discussion of the technical solutions and numerical results.

3.5.1. Theme 1: Energy Efficiency Strategies

Generally, almost all research under this theme provided retrofitting solutions utilizing field and experimental methods (case studies) and simulation tools.

Historical Contexts

The majority of these studies were accomplished in Italy; Piselli, Guastaveglia et al., and Piselli, Romanelli et al. retrofitted the infrastructure of historical buildings in Italy (offshore and educational buildings) within a European Project (GEOFIT Horizon 2020), namely with geothermal systems for cooling and heating [79,80]. They promoted utilizing emerging technologies, namely building information modeling (BIM), as a human-centered operational management approach supporting energy and environmental performance. Meanwhile, technologies with low architectural impact can preserve the cultural heritage identity.

Cellura et al. [68] retrofitted a rural building in a park in an ancient village in Italy to optimize the original materials and thermal comfort and reach the net zero energy target (reducing cooling, heating, and lighting loads). For this last aspect, they integrated active and passive solutions. The management authority was engaged in the early stages of decision making. To preserve the original building configurations, they renovated building systems, altering windows, internal insulation, and the renewable energy source (RES); however, the type of RES can be excluded due to visual reasons (i.e., wind turbine). The retrofitting solutions were restricted to national guidelines and legalization.

In this light, Caputo et al. [83] stated that the guideline does not provide adequately efficient solutions. In other words, "heritage preservation and energy efficiency measures are often conceived as mutually exclusive purpose" [63], regardless of the associated challenges of refurbishing protected buildings regarding architectural constraints. Menconi et al. [67] retrofitted dwellings using internal insulation to preserve the original features of the building, that are considered an asset of the historical landscape [53]. Eventually, Alev et al. [52] provided an interdisciplinary study to analyze renovation alternatives to improve the energy performance of historic rural dwellings in three Baltic Sea countries. The study revealed that older rural buildings have poor insulation and airtightness, $\cdots$, which require considerable upgrades. Among the various structural components, external wall insulation had the most significant energy-saving potential due to its large area and high thermal transmittance. With this, it is feasible to meet energy performance standards while maintaining the building's original appearance.

Agricultural, Remote, Tourist, and Mountainous Contexts

Muthuvel et al. [60] relied on technological solutions to achieve zero-energy homes by retrofitting dwelling building systems (home appliances) related to the residents' agricultural activities in a remote village. It found that the control is as simple as the system. For the tourism contexts, Congedo et al. [66] studied an adaptively reused farmhouse in Italy, by employing local construction materials and traditional practices to improve energy/environmental performance. This approach boosted the local economy of the cultural, tourism, and construction sectors, all as a valorization strategy to preserve traditional identity. They demonstrated how the building has been affected by the surrounding agri-economic activities (olive groves and vineyards) that add value by supporting agriculture tourism. Simultaneously, they highlighted the importance of linking the interventions with the top-down processes supporting Cellura et al.'s outcomes.

Sun and Leng [58] provided multi-objective optimization of rural dwellings in ethnic minority areas; these areas had a low standard of living and, consequently, poor energy performance of buildings. However, there are possible environmental benefits like less air pollution, higher atmospheric transparency, and solar radiation. From the relationship between thermal comfort and energy, Yanru Li et al. provided guidelines for refurbishing rural buildings to attain optimum indoor environment enhancement for elderly people [100]. In the studies that focused on optimizing these factors, it is inevitable to adapt retrofitting measures regarding different thermal comfort factors for occupants, especially for improving infrastructure, like energy sources, as mentioned by Cardoso and González: "thermal comfort levels in residential buildings not suitable for the children to perform school".

For industrial buildings, Udovichenko et al. [90] developed a design framework for retrofitting a building with a hybrid renewable energy-assisted hydroponics farming system to produce fresh food in a harsh climate in rural Canada. For this building type, the lead constraint is environmental factors (e.g., humidity and water), requiring suitable environmental control; in other words, improving building systems is an essential retrofitting solution.

In mountainous areas in China, J. Han and Yang [69] quantified the energy savings due to a passive retrofitting solution without demonstrating the relationship to or influence of the context, similar to what was implemented by Yanru Li et al. [100], who indicated that the air conditioning load was the most significant proportion of the total construction load. Limited land affects the building footprint and retrofitting.

Cao et al. [92] proposed energy-saving retrofit assessment techniques for natural ventilation because they noted that the existing research on energy-saving retrofit is aimed at air-conditioned buildings and is not suitable for rural dwellings. The authors concur with this observation as many papers have made this same assumption in different contexts [84,93,94,106].

In the same context, the authors discovered that many studies have concentrated on assessing energy conservation through retrofitting techniques for building systems or envelopes in certain situations. They employed simulation tools to evaluate the effectiveness of these methods, which resulted in improved energy efficiency, among other benefits. However, the impact of implementing these interventions in rural contexts was not thoroughly explained, as referenced in references [57,58,74,78,84,88,89,94,96,99]. In other words, the authors argue that these practices can be implemented in any context, particularly the social aspects that have not been explored. Thus, different approaches should be presented. Some of these studies conducted on-site surveys to calibrate the simulation model to minimize the discrepancy between simulated energy use and observed data, such as in references [68,88].

Other studies observed the same behavior but explained the relationship with the surrounding environment or with the building morphology itself, like the role of ventilation as a passive cooling measure with great potential for sustainable renovation of rural buildings [65]. Rural buildings can benefit from less air pollution, more daylight, and integration with the natural environment, namely shading with trees and wall vegetation [102], especially in forest-based settlements. Selecting and combining the renewable energy source (i.e., biomass) with a fuelwood source available due to the surrounding context for heating and or cooking activities [75,85] or selecting specific heating sources like the ground-source brine-to-water heat pumps are only suitable for rural areas because of land requirements [97]. Other studies discussed the impact of building typologies on energy consumption, such as detached dwellings having the largest heat loss because of the high surface area to volume ratio or construction typology of cavity walls allowing for more cavity insulation retrofitting [57,63], decides emphasizing the negative correlation between building age and readiness to retrofit.

### 3.5.2. Theme 2: Energy Efficiency Planning

Tahsildoost and Zomorodian [77] provided multi-criteria decision-making and defined optimum retrofit strategies for rural buildings based on their economic viability for four buildings in different climate zones in Iran. They highlighted the importance of government-supported programs and incentive methods to educate and subsidize rural buildings, especially in low-income areas, to appeal to the building owners.

In a Chinese context, Qi et al. [72] provided a geometric classification method to facilitate energy performance evaluation and stated the necessity to analyze the potential for energy savings and the effects of retrofitting measures. Liu et al. [110] evaluated potential energy savings, costs, greenhouse gas emission reductions, and adoption strategies for improving building envelope efficiency in rural residential buildings. Replacing current fuel subsidies with retrofit subsidies is a win–win–win for rural households, local governments, and the environment, as it is a more efficient approach, supporting Siudek et al.'s finding, which stated that easy and moderate retrofits could effectively reduce the operating heating costs of clean heating.

Cardoso and González [75] evaluated the impact of altering the energy source in rural Argentina. The shift from using fuelwood alone to a combination of energy sources (liquid petroleum gas and fuelwood) positively impacted the quality of life. This transition was also a part of a social assistance program to support vulnerable sectors. However, many households continue to rely on wood stoves for their high caloric power and the sense of security they provide, as they are not dependent on external sources of energy. This traditional practice remains prevalent.

In a European context, Zeyen et al. [97] discussed cost-effective ways to decrease space heating demand peaks; they found that retrofitting solutions and energy efficiency measures should be applied to manage thermal peak demands rather than reducing energy demand. In Poland, Ksiezopolski et al. [82] investigated the correlations between the thermal modernization of a rural single-family house (SFH) by altering the energy source from coal to a clean energy source and examining its impact on household income. It was found that the process of changing supply and consumption patterns would enhance energy and environmental security. Siudek et al.'s [86] investigated the correlations between the thermal modernization of SFHs, the changes in energy sources to clean energy, and the disposable income of households in Polish rural areas. It was found that switching from coal to alternative energy sources for new house construction can significantly increase the cost of building, especially when installing renewable equipment for heating and water.

Byrne and O'Regan [59] provided an approach to delivering relevant and accessible information to stakeholders, showing how community efforts have progressed in renewable energy production. The rural residents who rely on energy sources that produce more pollution (e.g., wood and heating oil) have a higher carbon footprint than that of urban residents. In Germany, Taruttis and Weber [103] explored the correlation between energy efficiency and the market value of SFHs to uncover the potential financial gains associated with investing in energy retrofits. Surplus for enhanced energy efficiency can attain a price that is superior to that in urban contexts. Rosin et al. [81] promoted a novel approach to modernizing the building envelope to maintain energy efficiency and biodiversity of species highly dependent on building structures. They advocated that retrofitting measures should preserve ecological values and mitigate adverse impacts on biodiversity.

### 3.5.3. Theme 3: Policy Evaluation

In the UK and Ireland, Ahern et al. [51] evaluated the energy retrofitting national scheme and realized characterization of rural dwelling typologies, energy efficiency, and carbon emissions. They utilized a market-based approach that found individuals can enhance the energy efficiency of their house without any upfront financial burden or added stress of debt obligations. The cost of retrofitting external insulation varied based on building size, insulation depth, and market conditions. Government involvement may be needed for exterior wall insulation retrofitting schemes, as it poses a financial risk to utility

companies. Walker et al. [52] examined the effectiveness of a national retrofitting scheme. They found that most retrofits are minor solutions that may not reduce fuel poverty, reflecting the importance of developing national schemes to tackle this. Collins and Curtis [61] evaluated the energy efficiency measurements of a national scheme, providing insights and recommendations for retrofitting strategies correlated with household characteristics.

The study of Žičkienė et al. [109] highlighted the significant political barriers in rural Lithuania as the local authorities are not directly involved in promoting RES usage, and there is insufficient support, exorbitant initial costs, and a lack of collaboration among stakeholders. In addition, they pointed out that the lack of adequate funding for research and innovation and sluggish academic research adversely affect the process. For these reasons, they engaged experts to provide policy implications, as they found that retrofitting public and residential buildings and energy infrastructure (e.g., energy-saving and controllable lighting systems) is more effective than switching energy generation from fossil fuels to RES as governments may turn a blind eye due to the high cost of RESs to achieve national energy security.

In Chinese contexts, references [54,56,71,76,105] have examined different approaches to assess building energy efficiency policies for retrofitting new and existing buildings and infrastructure to address challenges of clean energy reformation and energy efficiency. In general, in the study of He et al. in 2015, they stated that Chinese policies are usually more supportive of rural areas than urban ones [54]; in contrast, Huo et al. and T. Han et al. emphasized that the energy intensity of rural residential buildings has been doubled, because of national building energy efficiency projects.

Their findings indicated that rural policy development has largely been left behind at the expense of the welfare of significant household numbers. Meanwhile, the policies' implementation is too weak, and the applicable strategic plan remains unclear (despite the availability of provincial rural energy security guidelines [98]), especially with the difficulty of developing clean heating in more impoverished areas (with low-income residents). Thus, they emphasized the importance of adapting building energy efficiency policies and codes to consider the specific features of regions (especially for new buildings) like microclimate, house typology, ethnic characteristics, and ownership type. Moreover, there should be responsibility for financing and facilitating the market mechanisms managing long-term investments, and close coordination between various social departments to create awareness. Huo et al. [73] provided additional recommendations that the government should consider energy efficiency retrofitting plans according to floor space, and plan the construction period based on building stock (rural dwelling stock represents 40% of the total).

In Italy, Bosia and Savio [55] reviewed three regional cultural heritage guidelines; they showed that the guidelines provided an adequate diagnosis of the values and traditional features of the built environment to help the stakeholders better understand them. In addition, the policies focused on delivering local materials and traditional technologies, which is common in all Italian regions [66] (considering that Italian regulations exclude the listed buildings from the minimum energy requirements, even after retrofitting [68]); however, there is a lack of local materials and know-how. They showed the importance of having materials available and the building capacity, which require much effort due to the high population, which supports J. Li and Shui's findings [56].

### 3.5.4. Theme 4: Human Behavioral Aspects

Energy efficiency retrofit projects were investigated from the perspective of farmers and various stakeholders. In China, references [98,104,108] conducted three field studies in the households of several villages. The reasons that these rural Chinese residents were less enthusiastic and unwilling to retrofit can be summed up as the high initial cost, lack of appropriate fines and subsidies, sluggish promotion of the retrofitting program, and low local farmer acceptance; in addition, there may be a lack of knowledge about energy-saving technologies [71]. Meanwhile, the willingness to retrofit varied significantly due to various

housing characteristics and building typologies. Meanwhile, engaging all stakeholders is essential; the intention to participate of the government, energy-saving service enterprises, and village residents directly affect the promotion of energy-saving renovations of existing buildings. Top-down support remains pivotal by affording appropriate penalties and aid to help rural residents mobilize their passion for contributing to energy-saving.

According to Lai et al. [108], the condition of the resident's housing may impact their willingness to retrofit. While safety concerns may encourage retrofitting, individuals living in well-maintained farmhouses or detached villas may hesitate to do so. These findings contradict previous studies [95,105] that suggest that low income limits individuals' investment in improving building performance, as viewed from both bottom-up and political perspectives. The authors would agree with the notion that there is a negative correlation between income and the decision to retrofit. Collins and Curtis stated that those living in a modern building are more willing to invest in retrofitting than those in older ones.

A study conducted by Barry et al. [62] examined the impact of self-efficacy on energy usage in 647 households in the USA. The study found that individual habits and perceptions of energy significantly affected consumption. Additionally, occupant behavior plays a crucial role in reducing energy usage. In a study conducted in Poland by Kaya et al. [87], a different approach was taken that linked education level with the willingness to invest in retrofitting. The study found that residents with higher education levels showed a minor interest in retrofitting, whereas farmers showed a significantly lower interest in retrofitting compared to other economic activities. Furthermore, the study observed a negative correlation between family size and willingness to retrofit. Lastly, residents who purchased energy-efficient home appliances had a significantly higher probability of retrofitting compared to those who opted for exterior insulation.

## 4. Discussion and Conclusions

Implementing energy retrofitting methods is crucial for attaining sustainable development objectives and reducing the impact of climate change. Mainly, when working in rural areas, it is essential to take into account their specific characteristics: cultural and social aspects, economic patterns, and infrastructure challenges. Conversely, the complexity of energy retrofitting practices (the associated internal and external factors) sets them apart from other domains. That can be grouped into two main aspects: technical issues and issues related to the stakeholders themselves. These require an intensive focus from various standpoints and cross-disciplinarity collaboration to generate the knowledge that can help to effectively tackle and mitigate these uncertainties.

In this view, this study attempted to investigate the academic progress and trends and diagnose the activities for retrofitting the built environment of rural settlements (RQ1) beyond the numerical and quantified results and technical details. In addition to explore the factors that can influence these practices and their impact on the rural area's traditional identity (RQ2) and (RQ3), which was addressed through a systematic literature review.

### 4.1. Summary of Main Findings

The SLR used the keywords retrofitting, rural, and built environment. Afterward, 16 scholarly databases were searched for peer-reviewed articles published between 2012 and March 2023. The study was conducted; consequently, a restricted protocol was developed (Table 1). Application of the inclusion criteria resulted in a total number of 60 eligible studies. The search and analysis were conducted between March and May 2023, utilizing various search, screening, and data mining tools. The data were extracted and content analysis was conducted in line with the research problem; these aspects included bibliographical data (e.g., author, citation, journal, and affiliation), cross-disciplinarity among the collaborators, the macro context of implementation, micro-context rural patterns, analysis scope of articles, and thematic classification and focus. Then, a synthesis of the results was provided to address three research questions (Section 1).

A framework for addressing areas of concern related to retrofitting rural built environments was developed by classifying papers in a publication pool. The articles were classified based on their publication year and journal. The analysis revealed that the years from 2020 to 2022 saw the highest number of published papers, with a total of 11 and 10, respectively (almost doubled in 2017). This might reflect the increased research interest in this domain or it might have been because of the lockdown.

The citation number positively correlates with the publishing time; the older the article, the higher the number of citations. Google Scholar had the highest citation number for each paper, one-third more than those of the Scopus and Web of Science databases. The top-cited paper was J. Li and Shui's study [56] in 2015, which was 63% more than the other papers in the same year. The most frequent number of authors was four (15 times), and the highest number of authors (ten) occurred in two papers [80,101]; the average number of authors was 5–6 authors per paper. The total number of keywords was 297; the most frequent keywords were energy efficiency building, energy efficiency, energy consumption, renewable energy, China, retrofit and multi-objective optimization, sustainability, and rural residences.

Among the journals, *Energies* published the most papers with eight articles. At the same time, Elsevier published the highest number of publications, with 32 papers. The highest number of contributor affiliations was from China, with 28 affiliations, followed by Italy with 10. The most common affiliation was the University of Perugia, Italy (four times), followed by the Polish Academy of Sciences, Poland. Twelve affiliations were listed in two papers, and the remaining affiliation were distributed equally. Consequently, the majority of implementations were conducted in China, with 30 articles, and Italy with 9 articles. In terms of settlement pattern, in Italy, the majority of cultural heritage and historical publications had been conducted by seven papers. These facts might provide insights into the high academic interest of Chinese scholars in regenerating rural China.

Many papers provided multi-objective optimization approaches. Techno-economic analyses predominated to reduce energy and assess the feasibility. This was followed by the technical interventions to optimize energy efficiency and attain thermal comfort or reduce the associated GHG emissions, demonstrating the importance of providing an economic analysis of any interventions and emphasizing the positive correlation with thermal comfort. It is evident that the architecture and planning domain predominated as the most frequent domain. At the same time, the highest connectivity was among architecture planning with both environmental and applied science, whereas the energy domain had the most interactions with other domains (Figure 8).

### 4.2. Addressing Research Questions

The research was classified into four main themes: (1) applying energy efficiency strategies by promoting retrofitting measures with the aim of energy conservation, using field methods (e.g., case studies, questionnaires, and on-site monitoring), and employing simulation tools; the majority of studies were conducted from a bottom-up approach. (2) Planning for energy efficiency including providing guidelines and scenario planning, utilizing methods such as multi-criteria decision-making, econometric, analytical, and field, in addition to different tools like assessment tools, remote sensing, and simulations. Bottom-up and mixed approaches characterized these studies. (3) Evaluating and examining the content of existing retrofitting schemes or clean energy policies. (4) The papers investigating the social and behavioral aspects that can affect the willingness to retrofit.

Almost all the studies confirmed the low thermal conditions of rural buildings, which can be interpreted as the farmers and residents typically constructing their own houses using rural traditions and habits, resulting in envelope structures that do not meet energy-saving building design standards (the contemporary buildings' typologies are not exempted). Simulation tools are one of the most effective assessment tools for exploring the energy conservation potential of different retrofitting scenarios for better decision-making and pre-renovation planning for existing buildings. However, most studies only focused

on pre-retrofitting scenarios without accounting for calibrating the simulation results with in situ situations or providing a rational background behind applying specific retrofitting measures and the logical correlation with the intimate context; otherwise, it does not make sense to devote more retrofitting resources in this way.

Regarding this, a set of questions should be raised and investigated about the reliability of the obtained results; what if the same intervention is implemented in urban areas? In this case, it is enough to use simulation tools and alter weather conditions and locations. Therefore, the authors advocate for the essential need to provide more innovative/applicable solutions that cover techno-economic and socio-cultural aspects and develop these implementations into on-ground ones. In addition, there is a crucial need to monitor the impact of post-refurbishing. For example, when providing internal insulation to preserve the original features of an occupied dwelling, what is the acceptance rate of this solution, what is the time frame for implementation, and how will it affect the residents' daily life? Another question is, while providing policy implications like proposing retrofitting energy sources, to what extent will these recommendations be applied in line with the decision makers' priorities.

This can occur through collaborations between academic bodies, the private sector, experts, and practitioners, to obtain insightful findings and widen the perspective of the results. Conversely, how to engage the decision maker and the local community, which is a real challenge and ambitious goal, should be adequately investigated in order to align the various stakeholders on the same page and as active contributors. Hence, a vital question could be asked before any intervention: who are the right stakeholders, and what are the academic capabilities and qualifications of the research team to lead the retrofitting process? In particular, the high support by scientific and academic bodies and the essential role of energy efficiency scholars as knowledge brokers is needed to solve real-world problems. In the same context, the researchers should be open to inter- and trans-disciplinary collaboration.

In the same domain, it is evident that there is an absence of locals' engagement, apart from engaging the landlord in early decision-making. Hence, the residents should participate in retrofitting processes (at least be kept informed). They should be aware of the benefit of conserving energy and educated about energy efficiency through awareness campaigns via on-site visits, the media, or even social media, with particular reference to adjusting behavior for energy efficiency to reduce consumption patterns. In addition, the economic aspect makes them reluctant to invest in retrofit so we can alternatively show them how to apply simple and affordable techniques like altering home appliances to energy-efficient ones. Considering that building capacities are more complex in densely populated areas [55,56] and vice versa, these approaches are more effective in sparsely populated ones, like remote areas.

In general, policies vary by the different contexts, but the common finding was the vital role of decision makers and top-down development in fulfilling building energy efficiency. Energy code regulations for sustainable buildings should be updated with more environmental performance indicators concerning micro-context specificities (e.g., microclimate, economic activities, and building morphologies), providing long-term energy and carbon emission savings goals and an appropriate regulatory environment. The national guidelines should be considered during any bottom-up intervention and should to linked to the findings.

The decision making might not be energy-efficiency-oriented because of the national priorities that may focus on improving residents' quality of life by renovating the deteriorated rural dwellings and high poverty rates, as discussed in reference [112], or concentrating on the dwellings' visual appearance [78]. This does not preclude local authorities, for example, from employing cost-effective building retrofitting solutions rather than changing conventional energy sources, as mentioned in the Lithuanian case. Meanwhile, enabling the societal debate on domestic energy policies can support the whole scene. Promoting

building energy efficiency in rural areas is a daunting challenge that requires financing and training in building industry skills [56].

The micro-context provides constraints to retrofitting implementations; the visual aspects and original materials are essential when dealing with culturally significant buildings or historical rural dwellings. It is crucial to prioritize preserving their heritage value and exploring the surrounding activities and operations like management lighting systems/natural ventilation [67,79]. Improving building systems can offer practical solutions to address preserving historical value or replace the less visible envelope elements. This requires more investigations using emerging technologies. The external landscape is redesigned to respect internal activities and vice versa in enhancing the local identity. The authors argue that effectively engaging the locals to retrofit their built environment and putting them in the decision-making position can enhance the sense of belonging, promoting another aspect of settlement valorization.

Additionally, micro-contexts correlated with energy consumption patterns, such as the living habits of rural residents [54,62]. Different building shapes and orientations affect the context [71,72]. The type of renewable energy can be determined, such as relying on passive solar techniques with high solar irradiance, or likewise using wooden biomass in forest settlements. Clearly, the location affects or should affect the retrofitting implementations, supporting the study's hypothesis; this is, despite the obscurity of this part in many of the included articles, such as what has been discussed in Section 3.5.1.

To conclude, this research contributes to a better understanding of retrofitting practices in specific rural contexts beyond the numerical results of energy savings. It provided insights to interested parties, from broader perspectives for scholars who work in energy efficiency and sustainable rural development, in addition to local authorities, especially those dedicated to energy savings or historical heritage, to promote integrated retrofitting within energy communities' key concept in rural commons [7,113,114]. Moreover, the study provided a method for conducting SLRs, employing different tools to streamline the process that may be beneficial for early-stage researchers on a given issue. This review also offers crucial systematic details such as the authors' country and affiliations, data collection methods, methods, tools, analysis scope, journals and publishers, and citation numbers (presented in the Supplementary Editable Excel Files S2–S4). This information is essential for future researchers who work in retrofitting rural areas and serves as a valuable reference that could be developed further; therefore, scholars are highly encouraged to investigate other elements and provide additional insights and trajectories to regenerate the rural heritage. Contemporaneously, further investigations into the national action plans, providing innovative retrofitting solutions and methods to involve the right stakeholder wills be beneficial.

However, despite the SLR being based on a quantitative and qualitative examination, the authors declare that the restricted SLR protocol constitutes a limitation of the finding. The search terms focused on "retrofitting rural built environment," and altering it may affect the choice of the articles to be included in the research; this indicates that some articles were not included as they do not have the term "rural" in the title, abstract, keywords, or text, like what has been implemented in references [34,115]. Consequently, analogous studies can be accomplished utilizing a broader search strategy and considering different analysis scopes, which might lead to the finding of new themes.

**Supplementary Materials:** The following supporting information can be downloaded at: https://www.mdpi.com/article/10.3390/buildings13071586/s1, File S1: The exported RIS file that includes the 168 papers before applying the screening criteria. S2, S3, and S4 are presented in the same Excel file, where S2 is the analysis of the included SLR studies in line with the RQs, and S3 presents the analysis of authors and affiliations. S4 presents the supported table for analysis.

**Author Contributions:** Conceptualization, A.A.; methodology, A.A. and A.B.; software, A.A.; investigation, A.A.; resources, A.A.; writing—original draft preparation, A.A.; writing—review and editing, A.A. and A.B.; visualization, A.A.; supervision, A.B. All authors have read and agreed to the published version of the manuscript.

**Funding:** This research received no external funding.

**Data Availability Statement:** The data presented in this study are openly available in the Supplementary Files.

**Conflicts of Interest:** The authors declare no conflict of interest.

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
