# Peer review of "Insights and Evidence on Energy Retrofitting Practices in Rural Areas: Systematic Literature Review (2012–2023)"

_buildings, doi:10.3390/buildings13071586_

Round 1

Reviewer 1 Report

The paper aims to examine retrofitting practices in rural settlements worldwide in a decade to diagnose: the focuses that are being pursued, stakeholder engagement, and finally, the bi-correlation between rural contexts and interventions, and vice versa retrofitting contributions to valorizing the place identity.

Major comments:

- It was not clear from the article whether the working hypothesis was confirmed or not. Has the goal of the study been achieved or not? The article needs to be better structured according to the answers to the authors' three research questions.

- Authors should explain the rationality of the search strategy. Why was the search for publications limited to energy efficiency if this term is not mentioned in the purpose of the study?

- Authors should pay more attention to the recommendations of doi:10.1136/bmj.n71 and describe the methods and results of the study in more detail (for example, report the study's results in the abstract of the article).

Minor comments:

- Authors should use one Date axis option (e.g. Figure 5(a) and Figure 6(a)).

Author Response

Dear Reviewer 1, 
Thank you very much for your comments and for supporting the manuscript’s improvement. Here is the reply to your valuable comments:

Comment #001
- It was not clear from the article whether the working hypothesis was confirmed or not. Has the goal of the study been achieved or not? The article needs to be better structured according to the answers to the authors' three research questions.

Reply #001: 
The structure has been emphasized (the conclusion), and the hypothesis is clearly confirmed; consequently, heading titles have been added in the conclusion section (Section 4.1 and Section 4.2). 
Additionally, the introduction has been updated to better introduce the research questions.

Comment #02: 

- Authors should explain the rationality of the search strategy. Why was the search for publications limited to energy efficiency if this term is not mentioned in the purpose of the study?

Reply #002
We do agree with this point. Therefore, the title and the purpose have been modified: clearly indicating Energy Retrofitting. Also, in the introduction, the focus on energy efficiency issues has been emphasized. 

Comment #03: 

- Authors should pay more attention to the recommendations of doi:10.1136/bmj.n71 and describe the methods and results of the study in more detail (for example, report the study's results in the abstract of the article).

Reply #03: 
 The abstract has been revised based on PRISMA 2020 checklist. 
The obtained results, conclusions, and limitations have been significantly enhanced (Lines 21-31).

Reply #04:
The date axis option has been applied. The title of Figure 6a. has been updated.

Reviewer 2 Report

The manuscript is very interesting and prepared with lots of effort. My only comment is the lack of elaboration of the research questions. If read from the beginning, the expectation is that the review will probably assess the climate impact of rural buildings. The research questions come as a surprise. They should be better justified and integrated in the introduction and repeated in the discussion/conclusion section. This refers to all 3 RQs.

Minor:

Table 1 might be moved to the Appendix

In line 37 there is a missing full stop

Author Response

Dear Reviewer 2,
Thank you very much for your comments which will help improve our manuscript. Here is the reply to your valuable comments:

Comment #001:

The manuscript is very interesting and prepared with lots of effort. My only comment is the lack of elaboration of the research questions. If read from the beginning, the expectation is that the review will probably assess the climate impact of rural buildings. The research questions come as a surprise. They should be better justified and integrated in the introduction and repeated in the discussion/conclusion section. This refers to all 3 RQs.

Reply #001 
Thank you for the feedback it encouraged us. Actually, we do agree with your comments. To enhance the coherence of the manuscript:

- The introduction has been expanded, elaborating more on the nature of rural areas, namely their specificity focusing on their values, supporting this by references indicating the need to valorize them during development (including energy retrofitting). In addition to focusing on energy efficiency and retrofitting and aligning the intervention with sustainable development goals (goal 7), also the title and study's purpose have been updated to support this. All for a better introduction to the research questions. 

- The conclusion section has been enhanced, clearly indicating the answer to the 3 RQs and confirming the hypothesis. 

Comment #002: 

Table 1 might be moved to the Appendix

Reply #002:
We have already attached a supplementary Excel file that has all the bibliographic details. But, in order to keep the readers' focus on the manuscript's sequence, we would think that leaving the table in this sequence (showing the reference number and only the title in a one-page table) would enable them to screen the titles included papers and support our argument. 

Comment #003 
In line 37 there is a missing full stop

Reply #003
Thank you. The entire manuscript has been revised grammatically. 

Additionally, the abstract has been improved.  

Round 2

Reviewer 1 Report

 Accept in present form